# Visually-Grounded Library of Behaviors for Manipulating Diverse Objects across Diverse Configurations and Views

**Jingyun Yang**[*]    **Hsiao-Yu Tung**[*]    **Yunchu Zhang**[*]    **Gaurav Pathak**
**Ashwini Pokle**    **Christopher G Atkeson**    **Katerina Fragkiadaki**
Carnegie Mellon University
{jingyuny,htung,yunchuz,gauravp,apokle,cga,kfragki2}@andrew.cmu.edu

**Abstract:** We propose a visually-grounded library of behaviors approach for learning to manipulate diverse objects across varying initial and goal configurations and camera placements. Our key innovation is to disentangle the standard image-to-action mapping into two separate modules that use different types of perceptual input: (1) a behavior selector which conditions on *intrinsic* and *semantically-rich* object appearance features to select the behaviors that can successfully perform the desired tasks on the object in hand, and (2) a library of behaviors each of which conditions on *extrinsic* and *abstract* object properties, such as object location and pose, to predict actions to execute over time. The selector uses a semantically-rich 3D object feature representation extracted from images in a differential end-to-end manner. This representation is trained to be view-invariant and affordance-aware using self-supervision, by predicting varying views and successful object manipulations. We test our framework on pushing and grasping diverse objects in simulation as well as transporting rigid, granular, and liquid food ingredients in a real robot setup. Our model outperforms image-to-action mappings that do not factorize static and dynamic object properties. We further ablate the contribution of the selector's input and show the benefits of the proposed view-predictive, affordance-aware 3D visual object representations.

**Keywords:** Robot Learning, Visual Representation, Interactive Perception

## 1  Introduction

Object manipulation in unstructured environments is challenging since methods to manipulate objects largely depend on the object's visual appearance. One approach to capture the dependence between actions and visual features is to learn a direct mapping from image to actions with deep neural networks [1, 2, 3]. Despite their flexibility, such end-to-end image-to-action mappings have been shown to be data aggressive [4], and cannot easily generalize across objects, camera viewpoints, or scene configurations [5].

Approaches that abstract away object details and encode only a subset of their properties, e.g., their 3D locations and velocities [6] or 3D keypoints [7, 8] make the state-to-action mapping easier to learn with less data. However, this abstraction may substantially limit the range of objects that a policy can handle, since useful information (object shapes, softness, weight, and material, for example) for the downstream task may be ignored. The challenging question is: how can we design a framework for object manipulation that uses **abstract** representations for sample efficient behavior learning, but at the same time is capable of utilizing **semantically-rich** representations for handling diverse objects and views.

We propose Visually-grounded library of BEhaviors (V-BEs), a hierarchical framework for vision-based object manipulation. Our main contribution is that the two levels of our policy hierarchy use different visual representations. At the lower level of the hierarchy, a behavior library contains a set of distinct behaviors each of which operates on an **abstract** object state representation that captures

---

[*] Equal contribution.

5th Conference on Robot Learning (CoRL 2021), London, UK.

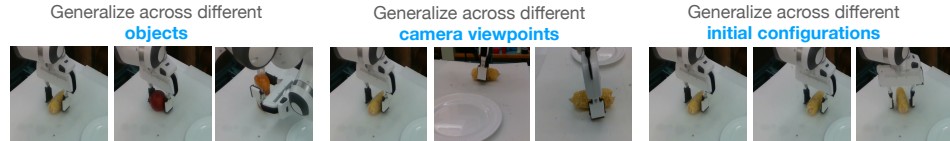

Figure 1: We propose a novel policy representation that generalizes to unseen objects, object placements and camera views.

dynamic properties of objects in the environment, such as object and gripper positions over time. Behaviors can be redundant, use different state abstractions, be closed loop or open loop, and be learned or manually engineered with different algorithms [9, 10, 11]. At the higher level of the hierarchy, a selector is trained using environment interactions to associate objects to behaviors that can successfully manipulate them. This selector takes as input RGB-D images of the scene, maps them to **semantically-rich** and view-invariant 3D object feature representations [12], and outputs the behavior that best suits the presented object and task. The decomposition of visual information into a perceptually rich selector network and behaviors with state abstraction makes the behaviors easier to learn with reinforcement learning or be designed by an engineer, and the selector easier to learn by trial-and-error learning as it does not bear the burden of producing low-level actions. At test time, we use 3D object detectors to localize the object, and present its location and size to the low-level controllers when necessary.

Our second contribution is proposing a visual representation for our behavior selector that is robust to changing camera viewpoint (with respect to a non-changing robot torso), as well as object placements. We use the differentiable 2D-to-3D neural networks of Tung et al. [12] to map an input RGB-D image alongside camera intrinsics and extrinsics to transform 3D visual scene feature maps from the current camera viewpoint to the world frame, ensuring that visual features are bound to a static world frame—that coincides with the frame of the agent's body—and do not change dramatically while the camera moves. The 3D visual feature representations learn based on environment interactions to embed objects close to the behaviors that can manipulate them. They are further trained from the auxiliary task of view prediction using a multi-view camera setup, which encourages the model to complete missing information from the current view so the 3D visual feature maps are consistent across views and through occlusions.

We test the proposed model in both simulated and real robot setups. In simulation, we consider two manipulation tasks: pushing and grasping. We show that our model can manipulate diverse objects and generalize the learned skills to unseen objects with varying object starting positions, initial poses, goal locations, and camera viewpoints. We show that our model outperforms existing end-to-end image-to-action mappings [4] and state abstracted object-to-action mappings that use only 3D object locations [6]. We ablate the contribution of our higher-level 3D feature representation used in our selector and then compare against 2D pretrained baselines. Lastly, we test our model in a real robot setup where the robot transports rigid, granular, and liquid objects onto a plate. We provide supplementary materials including additional qualitative results at https://yjy0625.github.io/projects/v-be/.

## 2 Related Work

**Libraries of Behaviors:** Libraries of behaviors have been considered in many previous works, and library elements have been called action, motion, behavioral, or motor primitives, parameterized policies, options, chunks, macros, subroutines, behavioral units, and skills, as well as many other terms. However, most previous work either assumes that the environment state is known and does not use perception, or has a different approach to perception than ours. Work of Neumann et al. on modular policies [13] defines a motor primitive as a mapping from a robot's joint angles and velocities and not the locations of objects. Rather, object locations are used as context for selecting a motor primitive. This means i) motor primitives are "blind" to the object 3D location throughout the episode, and ii) object appearance is discarded which means the library of primitives cannot handle object variability. Work of de Silva [14] on parameterized policies similarly considers as the contextual vector the desired goal location of a dart, and learns a mapping from this location to policy parameters, where each policy operates only over the robots' joints and is represented as a Dynamic Motion Primitive (DMP) [15, 16, 17]. To extend DMPs to tasks involving physical

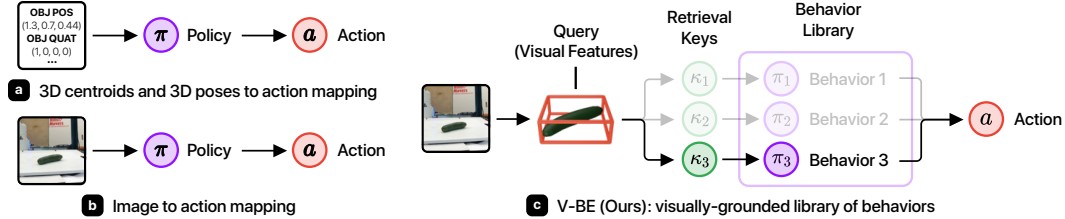

Figure 2: Comparison of our model with previous approaches. In contrast to state-to-action or image-to-action mapping, the proposed framework decomposes a policy into a behavior selection module and a library of behaviors to select from. The decomposition enables these modules to work on different representation: the selection module operates in a semantically-rich visual feature space, while the behaviors operates in an abstract object state space that facilitate efficient policy learning.

interactions, Kroemer et al. [18] parameterize DMPs based on the distance of an object part from the gripper, but only consider known object 3D shapes. Strudel et al. [19] use a depth frame as input to a meta-policy that selects skills to execute. However, the use of modularity over skills is for temporal sequencing: there is no mixture of behaviors per manipulation task, rather an image to action mapping is learned with behavior cloning from demonstrations.

**Deep Reinforcement and Imitation Learning for Object Manipulation:** Current work in (deep) reinforcement learning typically considers only one monolithic policy learned via reinforcement learning [9, 20, 21], imitation learning [22, 23, 1], or a combination of the two [24, 25]. Most approaches take as input 3D object and gripper centroids, orientations, and velocities, and ignore contextual visual information [26]. Existing methods are commonly trained in a fixed environment and evaluated in terms of their ability to discover a policy rather than their ability to generalize to previously unseen circumstances, e.g., novel objects and views, which is the goal of this work.

Methods that do aim to generalize across objects learn a mapping from images-to-actions [4], depth-to-actions [27] or pointcloud-to-actions [28, 29]. They have been successful in various tasks including object grasping [27, 30] and object pushing from a fixed camera view [31, 4]. Seminal works of [4, 32] train an image-to-action deep neural network by imitating trajectories obtained from planners that operate in a low-dimensional state space. Nevertheless, such monolithic image-to-action deep mappings don't generalize well across different camera viewpoints [5, 33, 34] and objects. Works that attempt to transfer visuomotor policies learned in simulation to the real-world often require identical placement of the camera in the real world [35, 6]. Florence et al. [36] train manipulation policies parameterized by the 2D or 3D locations of a designated set of visual descriptors obtained from RGB-D images, and show their policies generalize across objects of the same object category, e.g., shoes of different shapes. The proposed work further allows generalization across object categories, and no explicit descriptor selection or optimization is necessary. Furthermore, our framework permits each behavior to use a different state representation.

**Hierarchical Reinforcement Learning (HRL):** Most existing HRL methods [37, 38] focus on discovering behaviors and temporally sequencing them for long-horizon tasks. This paper instead uses a hierarchical vision-based policy architecture to improve model generalization across objects, configurations, and viewpoints.

Please check supplementary materials for additional discussion of related work.

## 3 Methods

**Problem Setup** We consider the problem of manipulating (e.g. grasping, pushing, etc.) diverse sets of objects from various initial configurations and camera viewpoints. At training time, the agent has access to training objects $\mathcal{O}_{\text{train}}$, RGB-D images from $J$ viewpoints $v_1, \ldots, v_J \in \mathcal{V}$, and it can interact with the training object from randomly selected viewpoints. At test time, the agent has to perform the same manipulation task as at training time on test objects $\mathcal{O}_{\text{test}}$ from selected viewpoints. Performance is measured by the success rate of manipulating all test objects with random initial configurations and camera viewpoints.

Our framework is comprised of two major components — (1) a library of behaviors $\Pi = \{\pi_i \mid i = 1, 2, \ldots, K\}$ and (2) a selector function G. Given an RGB-D image $I$ of the scene captured from camera view $v$, denoted by $I_v = \{I, v\}$, and an object 3D bounding box o, the selector $G$ obtains the

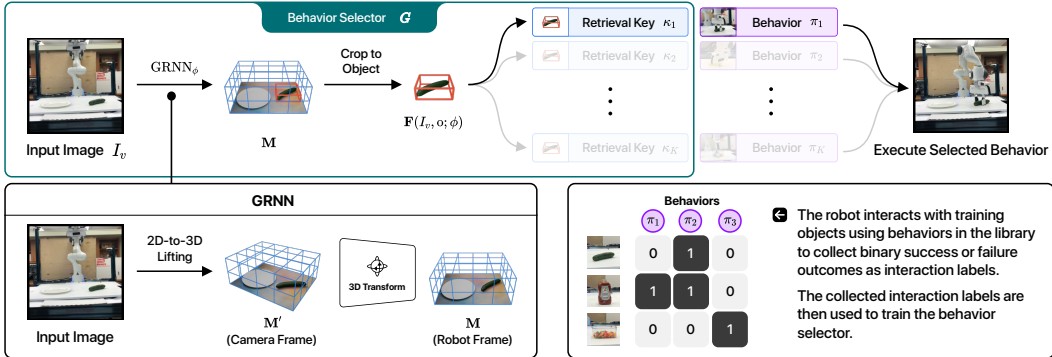

Figure 3: Overview of the proposed framework.

probability of successfully manipulating the object when applying behavior $\pi_i$ on the object by computing a query object feature representation $\mathbf{F}(I_v, \mathrm{o})$ and compares it with learned key embeddings $\kappa_i = \kappa(\pi_i)$, $i = 1, ..., K$ associated with each behavior $\pi_i$,

$$\mathrm{G}(I_v, \mathrm{o}, \pi_i; \phi, \kappa) = \sigma(\langle \mathbf{F}(I_v, \mathrm{o}; \phi), \kappa_i \rangle) \in [0, 1], \tag{1}$$

where $\phi$ is the learnable neural network parameters for function $\mathbf{F}$, $\langle \cdot, \cdot \rangle$ is the inner product operation and $\sigma$ is the sigmoid function. We will detail the feature dimension of $\mathbf{F}(I_v, \mathrm{o}; \phi)$ and $\kappa$ in the next section. At test time, we select the behavior $\pi_i$ that has the highest probability of leading to successful manipulation.

Each behavior $\pi_i$ uses its own abstract state representation for the objects to manipulate, for example, 3D object locations and 3D poses, and does not necessarily take into account other aspects of the input image. Each behavior is designed or learns to handle only a subset of object shapes and orientations. To handle a diverse set of objects and orientations, our selector learns to pick different behaviors for different objects or orientations. One advantage of the modularity of our policy architecture is that each behavior can use its own state abstraction, such as object 6D-poses, 3D bounding boxes, part-based 3D boxes, or 2D or 3D object keypoint locations [7]. Our framework supports integrating a wide range of existing models, representations, and controllers as selectable behaviors.

In the rest of this section, we detail the architecture and training details of the behavior selector in Section 3.1, and describe how we acquire a library of behaviors for grasping and pushing—the two robot manipulation tasks we evaluate our framework on—in Section 3.2.

### 3.1 Visually-Grounded Behavior Selector (G)

The behavior selector G is a classifier that learns object-centric visual feature representation $\mathbf{F}(I_v, \mathrm{o}; \phi)$ for the object box $\mathrm{o}$ in image view $I_v$, and behavioral key embeddings $\kappa^i$ ($i = 1, \ldots, K$) for the behaviors in the library to retrieve behaviors compatible with a particular object.

**Training the selector with interaction labels.** We learn the object feature representation and behavioral keys through trial-and-error. In each trial, our agent applies a randomly sampled behavior $\pi_i$ on an object $\mathrm{o}$ in the workspace which results in binary success or failure outcome $\ell \in \{0, 1\}$, which we call *interaction labels*. Agent interactions are organized as tuples of the form $(I_v, \mathrm{o}, \pi_i, \ell)$. With the interaction experience as training data $\mathcal{T}$, we train the feature encoder $f$ and behavioral keys $\kappa_i = \kappa(\pi_i)$ ($i = 1, \ldots, K$) with the loss:

$$\mathcal{L}_{\mathrm{afford}}(\kappa, \phi) = \sum_{(I_v, \mathrm{o}, \pi_i, \ell) \in \mathcal{T}} \mathrm{BCE}(\ell, \mathrm{G}(I_v, \mathrm{o}, \pi_i; \phi, \kappa)) = \sum_{(I_v, \mathrm{o}, \pi_i, \ell) \in \mathcal{T}} \mathrm{BCE}(\ell, \sigma(\langle \mathbf{F}(I_v, \mathrm{o}; \phi), \kappa_i \rangle)) \tag{2}$$

where $\mathrm{BCE}(y, p) = -y \cdot \log(p) - (1 - y) \log(1 - p)$ is the binary cross entropy loss.

**3D object feature representation.** Our object feature representation $\mathbf{F}(I_v, \mathrm{o}; \phi)$ is computed using Geometry-aware Recurrent Neural Networks (GRNNs) of Tung et al. [12], which are end-to-end differentiable architectures that map a single RGB-D image or a set of multi-view images to 3D feature representation of the scene the image(s) depict. Given a posed image $I_v$, GRNNs obtain scene-centric 3D feature representation $\mathbf{M} = \mathrm{GRNN}_\phi(I_v) \in \mathbb{R}^{H \times W \times D \times C}$ through differentiable

2D-to-3D operations, 3D convolution-based refinement, and 3D rotation operations that align the 3D feature representations with the robot's coordinate frame (as opposed to camera frame), with $W, H, D$ denoting the width, height and depth of the scene map and $C$ denotes the feature dimension of the 3D scene feature map.

From the scene map $\mathbf{M}$, we obtain object-centric feature representation $\mathbf{F}(I_v, \mathrm{o}; \phi) = \mathrm{crop}(\mathbf{M}, \mathrm{o}) \in \mathbb{R}^{64 \times 64 \times 64 \times 32}$ by cropping the scene map using a fixed-size axis-aligned box $\mathrm{o}$, centered around the object we wish to manipulate. The feature cropping operation is similar to the one used in Mask-RCNN [39]. The retrieval policies keys are also learned in the same representation space. Thus, both $\mathbf{F}(I_v, \mathrm{o}; \phi)$ and $\kappa_i, (i = 1, \dots, K)$ have length $64 \times 64 \times 64 \times 32$ in the experiments.

**3D object detector.** We learn a detector with 3D Mask-RCNN built on top of the GRNNs feature encoder [12]. We use groundtruth 3D object boxes at training time, and predicted 3D object boxes at test time, where we train our representation to detect objects in 3D.

**View prediction and occupancy prediction as an auxiliary task.** We use view prediction and occupancy prediction as an auxiliary task to help our image encoder generalize better in its ability to select behaviors. These two self-supervised prediction tasks have been shown to provide a useful pretraining or co-training objective for 3D object detection in [40]. Given an input posed image $I_v^n$ and a query view $q^n$, the overall self-supervised prediction loss reads:

$$\mathcal{L}_{\text{self-pred}}(\phi, \theta, \eta) = \sum_{n=1}^{N} \underbrace{\|P_\theta(\mathrm{GRNN}_\phi(I_v^n), q^n) - \bar{I}_q^n\|_2^2}_{\text{view prediction loss}} + \underbrace{\|\mathrm{Occ}_\eta(\mathrm{GRNN}_\phi(I_v^n)) - \mathrm{occ}^n\|_1}_{\text{occupancy prediction loss}}, \quad (3)$$

where $P_\theta(\mathbf{M}, q)$ is a projection function that projects a 3D feature map $\mathbf{M}$ from the query viewpoint $q$ to a 2D feature map and decodes it to a target image $\bar{I}_q^n$ using an image decoder with neural network weights $\theta$, $\mathrm{Occ}_\eta(\mathbf{M}) \in \mathbb{R}^{64 \times 64 \times 64}$ is a voxel occupancy prediction function that predicts a 3D occupancy map from an input 3D feature map $\mathbf{M}$ using a single 3D convolution layer with weights $\eta$, and $\mathrm{occ}^n$ is the estimated occupancy map computed from all available input views in the $n^{\text{th}}$ data point by voxelizing the unprojected point clouds from all available depth images. We train the model from unlabelled multi-view images captured around the table by simply moving the cameras, capturing the images, and recording the corresponding camera locations.

The final objective for training the affordance-based visual features is

$$\underset{\kappa, \phi, \theta, \eta}{\text{minimize}} \, \mathcal{L}(\kappa, \phi, \theta, \eta) = \mathcal{L}_{\text{self-pred}}(\phi, \theta, \eta) + \lambda_\text{a} \cdot \mathcal{L}_{\text{afford}}(\kappa, \phi), \quad (4)$$

where $\lambda_\text{a}$ is a hyperparameter for balancing the two losses.

## 3.2 Building a Library of State Abstracted Behaviors

Any existing behaviors, whether engineered or learned using reinforcement or imitation learning, can be included in our library. This flexibility is a contribution of our modular architecture. In this paper, we consider three common manipulation tasks: pushing, grasping, and transporting. We build appropriate behavior libraries for each.

In pushing, the behaviors are deterministic goal conditioned policies $a_t = \pi(s_t, g)$ that map a state of the environment and the robot $s_t = [s_t^e, s_t^r]$ and a goal state $g$ to an action $a_t$ at time step $t$. The environment state $s_t^e$ is the 3D object centroid and the robot state $s_t^r$ is the gripper 3D location, pose, and whether it is opened or closed. Actions include 3D translation, opening (position control), and closing (force control) of the gripper. A goal state $g$ is a target centroid location for the object. We use a total of 25 goal conditioned policies – one is trained from the whole set of objects, while the others are trained on disjoint subsets of object configurations organized based on object category and initial poses. We train all policies using deterministic policy gradients (DDPG) [25] with goal relabelling (HER) [9] while randomizing initial and goal object 3D locations.

In grasping, we design controllers $\pi(a_t | g; p^{grasp}, q^{grasp})$ which given a 3D grasping point $p^{grasp} \in \mathbb{R}^3$ relative to the center of the object and a grasping 3D angle $q^{grasp} \in \mathbb{R}^2$, move the gripper (open loop) to the grasping 3D point location, close it, and move it to the desired goal location. The grasping angle $q^{grasp}$ consists of two numbers describing the yaw of the gripper and the elevation angles between the gripper and the table surface. When the elevation angle is smaller than 90 degrees (not top-down grasps), we constrain the gripper to point toward the center of the object on

the x-y plane. We manually select 30 different controllers including top-down grasps with different yaw orientations (top-grasps) and grasps from the side with different elevation angles of the gripper (side-grasps). We empirically found that these parameterized controllers are quite stable and can be shared across multiple objects. More details are provided in the supplementary materials.

## 4 Experiments

Our experiments aim to answer the following questions: (1) Does the proposed library-based approach outperform existing methods that use a single combined perception and policy module, either using 2D images, 3D object locations, or 3D scene feature maps as input? (2) Is the proposed view-invariant and affordance-aware 3D feature representation a necessary choice for the selector? (3) Does the method work on a real robot? We test our model on grasping and pushing a wide variety of objects in the MuJoCo simulator [41] and further test a transporting task on a real-world Franka Panda robot arm.

### 4.1 Simulation Experiments

Our simulated environment consists of a Fetch Robot equipped with a parallel-jaw gripper. The robot is positioned in front of a table of height $0.4m$. To obtain the visual observations, on each episode we choose 3 random cameras from cameras placed at 30 nominal different views including 10 different azimuths ranging from $0°$ to $360°$ combined with 3 different elevation angles from $20°$, $40°$, $60°$. All cameras are looking at the center of the table top, and are 0.5 meter away from that point. All images have size $128 \times 128$.

**Task Descriptions:** In the grasping task, the agent has to grasp an object and move it to a specified target location above the table. We use 274 distinct object meshes from 6 categories in ShapeNet [42] including toy buses, toy cars, cans, bowls, plates, and bottles. The materials and densities of all objects are identical. We randomly split the dataset into 207 training objects, and 67 testing objects. After augmenting the meshes with random scaling from 0.8 to 1.5 and random rotations around the vertical z-axis, we get a total of 800 distinct object configurations (object instance and pose), 600 for training and 200 for testing. At the start of each episode, an object is placed in an area of $30cm \times 16cm$ around the center of the table, and a goal is sampled uniformly $10 \sim 30cm$ away from the gripper's initial position. An episode is successful if the object centroid is within $5cm$ of the target at the final timestep.

In the pushing task, the agent has to push an object placed on the table to a specified target location. We use 100 objects from 12 categories in ShapeNet [42]: baskets, bowls, bottles, toy buses, cameras, cans, caps, toy cars, earphones, keyboards, knives, and mugs. After augmentation and splitting to train and test sets, we obtain 615 training object configurations and 200 for testing. The initial and the goal position of the object are both uniformly sampled to be within $15cm$ of the center of the table along both x-axis and y-axis, although we resample if that location is already in the goal area. An episode is successful if the object centroid is within $5cm$ of the goal within 50 timesteps.

**Baselines:** We compare our method with various learning and non-learning based methods for object manipulation:

(a) *Single Behavior w/ Abstract 3D State (Abstract 3D) [9, 25]:* a policy takes as input ground truth 3D bounding box of the object and gripper and outputs actions.

(b) *Single Behavior w/ Abstract 3D State and 2D Images (Abstract 3D + Image):* a policy takes both RGB-D images and the ground truth 3D bounding box as inputs and outputs actions. Our architecture resembles that of [43], but we further include ground truth object position as extra inputs to the model. For fair comparisons to other methods, the model only takes as input the current state as opposed to the states in 5 past steps, as in [43].

(c) *Single Behavior w/ 3D Feature Tensor (Contextual 3D):* a policy takes as input RGB-D images and the ground truth 3D bounding box and outputs actions. Different from (b), the model first transforms the image into a view-invariant 3D feature tensor using GRNNs [12], then converts the 3D feature tensor into a feature vector though three 3D-convolutional layers and a fully connected layer, and concatenates it with the rest of the inputs to predict actions.

(d) *Ours, Library of Behaviors w/ Visual Selector (V-BEs):* Our model takes the same input as (b) and (c). The 3D bounding boxes are used as input to all the behaviors. The RGB-D images are transformed into 3D affordance-aware visual features and treated as input to the selector.

| | Single Behavior | | | Library of Behaviors | Ablation study on the selector's visual feature representation | | |
|---|---|---|---|---|---|---|---|
| | Abstract 3D [9, 25] | Abstract 3D + Image | Contextual 3D | V-BEs (Ours) | V-BEs w/ 2D features | V-BEs w/o Fine-tuning on Interaction Labels [45, 12] | |
| grasping | 0.30 | 0.35 | 0.20 | **0.78** | 0.46 | 0.31 | |
| pushing | 0.83 | 0.70 | 0.10 | **0.88** | 0.81 | 0.46 | |

Table 1: Success rates on grasping and pushing unseen objects. We also ablate the proposed method with selectors operating on varying representations.

We train the baselines with different learning methods including behavior cloning [4], DDPG-HER [11, 9] and DAGGER [44]. We report the best performance we got by training the model with these different methods. We also attempt to make all the models have similar number of parameters so the comparison is fair. However, larger networks are empirically harder to train and do not converge well, so we instead increase the number of parameters in smaller networks until their performance saturates. For pushing, we found that using DDPG-HER is enough to lean a good *Abstract 3D* policy from scratch. For *abstract 3D + Image*, we found it is critical to use behavior cloning from expert demonstrations to obtain good policies. The expert demonstrations are obtained from trained expert policies on single objects. For *Contextual 3D*, we include DAGGER to enforce behavior cloning during execution. To train the grasping policies, we further include human demonstrations in the replay buffer when training it with DDPG-HER. Both *abstract 3D + Image* and *Contextual 3D* are trained with DAGGER since offline behavior cloning is insufficient.

## 4.2 Single Behavior versus a Library of Behaviors

We compare the proposed model with models that do not use a library-based approach, i.e., single behavior approaches. As shown in Table 1, our method outperforms all the single behavior baselines. *Abstract 3D* performs well, but since it does not use any visual information, its performance saturates at around 0.8 for pushing and 0.3 for grasping. *Abstract 3D* performs poorly for grasping. The learned behaviors do not transfer well to new objects. Adding a 2D image helps, but not dramatically (see *Abstract 3D + Image* in Table 1). Although 3D feature maps obtained from GRNNs are semantically rich and can handle varying viewpoints, the mapping to actions is harder to learn due to the higher dimensionality of the 3D scene map, resulting in under-fitting models. Our model takes advantage of both abstract and semantically rich representation and thus can handle better object variability and transferability. The combinatorial nature of the proposed method allows the model to capture the multi-modality in trajectory generation.

## 4.3 The Necessity of Building the Selector with the Proposed 3D Representations

Next, we show the importance of using view-invariant 3D visual feature representations and fine-tuning the selector with interaction labels. We compare our method with two baselines: (a) a model with a selector that learns the visual affordance features using 2D visual features extracted from 2D CNNs, and (a) a model with a selector that operates over 3D visual feature representation learned only with the view and occupancy prediction loss, as suggested in [45, 12], without fine tuning with interaction labels. See Table 1 for the results. Our method significantly outperforms these two baselines, which shows the importance of both proposed components. To fully test the power of existing 2D CNNs, we also tested 2D feature selector with existing VGG network [46] pretrained on ImageNet and fine-tuned on our interaction labels. However, the performance ( a success rate of 0.78 on pushing) does not differ too much with shallower 2D CNNs trained from scratch.

## 4.4 Transporting Task on a Real Robot

We test our model on a 7-DOF Franka robot arm equipped with a parallel-jaw gripper (using [47]'s software stack) for a transporting task, where the robot needs to transport various rigid, granular, or liquid food ingredients from random initial positions and poses onto a plate (see Figure 4). We set up 4 Intel RealSense RGB-D cameras that have full view of the workspace around the the center of the table. In each trial, an object is placed in a $50\text{cm} \times 30\text{cm}$ region on the table, and the goal is transport all objects to a plate 25cm to the left of the starting region. Granular objects and liquids are placed inside containers in the beginning of the trial. An episode is considered successful if the object is successfully transported into the plate. For granular objects and liquid, an episode is considered successful if at least half of the total quantity ends up in the plate.

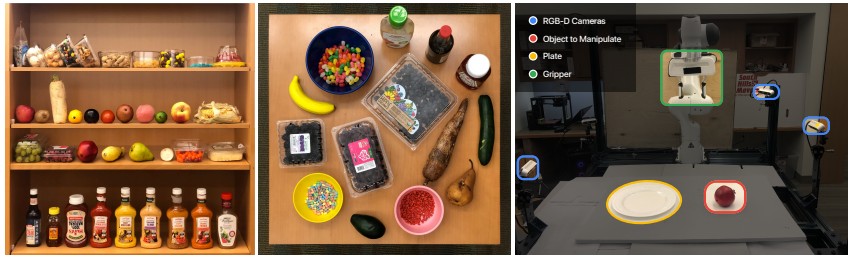

|Training Objects|Testing Objects|Workspace Setup|

Figure 4: Real robot experimental setup. We set 4 cameras around the table that cover different viewpoints of the workspace. 3 out of 4 cameras are shown in the right most image. An extra topdown camera is used, but it is clipped from the image.

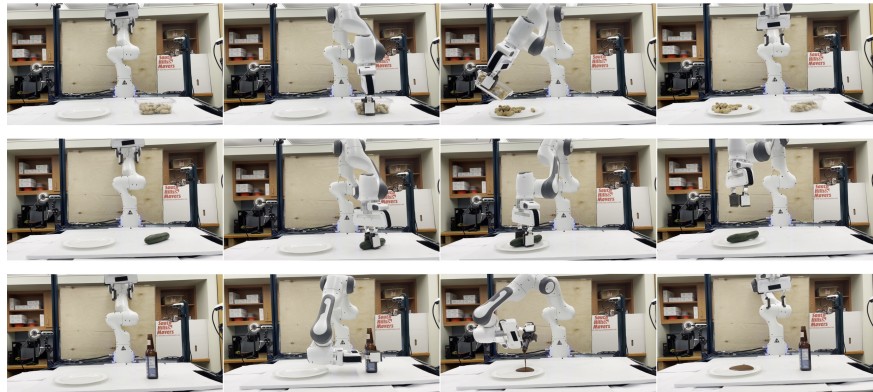

Figure 5: Sample real robot executions. Each row shows an execution trial during test time. Our robot can successfully transport rigid, granular, and liquid food ingredients to a target plate.

We construct our library of behaviors with 26 controllers, in which 13 of them are pick-and-place controllers from various grasping angles and the other 13 are pick-and-pour controllers using the same grasping angles as the pick-and-place ones. At evaluation, we randomly select 3 views from 4 possible camera views to obtain RGB-D images as inputs to the learned selector. We use 20 rigid objects, 20 granular objects, and 12 bottles of liquids for our experiment, and we split them into 38 training object and 14 testing objects. To train our model, we collect a total of $3510$ interaction labels by running the 26 behaviors on the training objects with random initial poses and using the labels to finetune the visual selector with the objective specified in Equation (4).

Our model achieves a success rate of **88.6%** on the test set. We compare our model with two baselines: 1) an image-to-action model trained with behavior cloning (*Image*), and 2) a hierarchical model that uses a library of behaviors and a selector with 2D representations (*V-BEs with 2D features*). We use the data collected during the interaction label collecting process as the data used to train both baselines. For the *Image* baseline, we are not able to get it to work at all, while we did make it work in simulation where there is more data. It may need more data to learn a general and robust policy in the real world [48]. For the *V-BEs with 2D features* baseline, we get a success rate of **38.0%** which is worse than the proposed model. This again shows the importance of operating the selector in the proposed view-invariant and object-centric 3D feature space. Sample executions and transporting objects in clutter are visualized in Figure 5 and included in our supplementary video.

## 5   Conclusion

We have presented V-BEs, a hierarchical policy architecture where 3D object visual feature representations are used to select from a library of behaviors. The proposed modular architecture supports both low-level behaviors and the selector to be learnt in a data efficient manner. We have shown results on pushing and grasping diverse objects in simulation and in the real world, across diverse viewpoints. Our method outperforms image-to-action monolithic policies of previous works, as well as policies that operate on 3D locations and velocities alone. Since our framework is sample-efficient and simple to run, we can easily deploy complex transporting skills on a real robot arm.

**Acknowledgments**

This work is supported by Sony AI, NSF award No 1849287, DARPA Machine Common Sense, an Amazon faculty award, and an NSF CAREER award.

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
