# OpenReview forum: "Visually-Grounded Library of Behaviors for Manipulating Diverse Objects across Diverse Configurations and Views"
_robot-learning.org/CoRL/2021/Conference — CoRL2021 Poster_

### Official Review · Reviewer_XqcZ · 2021-07-19

**Originality:** Fair
**Technical Quality:** Good
**Clarity Of Presentation:** Very Good
**Impact:** 3

**Recommendation:**

Weak Reject: I recommend rejecting the paper, but will not argue for my recommendation if the majority of other reviewers have a different opinion.

**Summary:**


Within manipulation tasks (this work focuses on grasping and adjacent skills), different behaviors are needed to generalize to different kinds of objects and their various orientations - for instance, pinching a container by its lip is specific to a certain family of objects but not others. While in principle it should be possible to train a single pixel-to-actions, goal-conditioned policy to subsume all these kinds of perceptual and control generalization, the authors opt to push the challenge of generalization to the perception stack only. Given some perceived object representation, and a fixed set of open-loop grasping “scripts” with different angles and grasp elevation angles, select the script most likely to succeed at the task. Concretely, RGB-D image goes into a GRNN, which creates a view-invariant 3D scene representation. The object-centric features are “cropped” from the 3D scene representation, which are the same feature space as the retrieval policies (64x64x64x32). The alignment between the object centric 3D representation and policy-specific key results in a fixed open loop pattern being played out.

An HRL interpretation of this work is that it’s an options policy for a 1-step MDP that learns which of N pre-defined goal-conditioned behaviors to select. The authors show that decoupling perception and planning in this manner leads to better performance compared to an end-to-end perception and policy, and also that their choice of object +behavior representation is superior to 2D images, 3D object locations, and 3D scene feature maps. Generalization to novel objects, novel viewpoints, and object poses is demonstrated on a real robot.

This is essentially an extension of Redmon and Angelova 2014 to 3D space and multiple viewpoints.


**Issues:**

I'd like the authors to comment on how this proposed framework is any different from simply parameterizing a K-way classifier for memorized trajectories, where the classifier is parameterized using a dot product on a learned object representation layer (but otherwise equivalent to a classifier).



**Reviewer Expertise:**

Good: General knowledge of the area

**Strengths And Weaknesses:**


Strengths
Policy is able to generalize to new objects and poses, from different viewpoints, on tasks that would be quite challenging for an end-to-end system to learn
Authors demonstrate that alternate object representations do not generalize as well to varying viewpoints and object configurations.

Weaknesses
Although the authors demonstrate that their choice of feature representation generalizes better than a naive 2D image representation, the downstream behaviors are very simple, scripted behaviors that are hand-tuned such that succeeding at the task means classifying one of K classes correctly.
The “library” is not really a library of different skills, but rather one or two behaviors with a discretized set of yaw and grasp elevation parameters, which drastically limits the effective DoF of behaviors to small, discrete action spaces. As mentioned in the summary, I believe the closest relevant work to this is not multi-task or options or even skills learning, but just visual grasp point detection generalized to 3D, as the learned “classifier” is just discriminating between candidate grasp poses.
State-of-the-art closed-loop grasping behavior from 2D images is already able to generalize to multi-view (Sadeghi et al 2017), novel objects and orientations (Kalashnikov '18). Given that the proposed work is essentially an open-loop grasping system (much, much easier and less general in regards to the skills that can be learned), I would have liked to see stronger generalization capabilities in the library.

**Summary Of Recommendation:**

The overall framework is quite interesting - pushing the problem of generalization as much as possible into perception + a standard classification problem (embedding alignment), and one that can be learned via hindsight data collection with a library of skills. I would have voted to accept this if this framework were applied to a more challenging library of skills, but instead as applied, there is *no* library skill learning on the real robot - it is merely just applying a classifier to a variety of scripted motions (with specific grasp angles and heights such that the classifier merely needs to pick the right grasp pose). The choice of the term "selector" sort of obfuscates this, but at the end of the day, it's really just a classifier over a set of memorized trajectories. I think the "selector" terminology would be more appropriate if the system were able to generalize in that "key" space, i.e. select new skills from embeddings that were not used at training time.

There is some novelty in the investigation of which 3D object representations computed from RGB-D are most ideal for predicting the "options policy",

---

> ### Author Response · Authors · 2021-08-30
> **Response to Reviewer XqcZ**
>
> We appreciate your valuable comments. It seems the major concerns are regarding comparison to previous work, and limitation in the current grasping behaviors. Below we address your concern. We have also updated the paper according to your suggestions (bold faced in the reply). Please do not hesitate to let us know if you have further concerns. We hope that we can convince you our key idea is valuable despite the simplicity of the method.
>
> > Comparison to K-way classifiers
>
> Yes, the proposed selector can be viewed as a learned classifier over skills, and indeed many similar works in robotics can be viewed as classifiers operating in discrete space or continuous space [1, 2, 3, 4, 5, 6]. **We have included an extra section on related work (Supplementary C.4) for the comparison.** To improve performance and generalization, the critical thing is to choose the right representation space for the classifier to operate on. Our selector works in a representation space that is object-centric, and view-invariant, and that is how it can generalize across object appearances, shapes, initial/goal configurations and camera viewpoints. We have empirically shown that naively using 2D representation from 2D CNN layers, which are widely applied in existing work, does not achieve as good performance.
>
> > Comparison to multi-task or skill learning works
>
> Thank you for your suggestion, **we have included this line of work in our extended related work discussion  in Supplementary C.3**. Existing works in multi-task or skill learning from image inputs [1, 5, 7, 8, 9] have shown impressive results in learning behaviors from demonstrations or interactions, but they often assume image inputs from a fixed camera view. [10] shows their reaching behavior can generalize to varying views by learning from multi-view data.  Our work aims to advance the solution for learning general manipulation skills that can handle various object types, shapes, visuals, initial configurations, and camera viewpoints . Our proposal is to combine readily available behaviors, instead of learning policies from scratch, and to advance the visual representation so the resulting model can be invariant to viewpoint change. We have put effort into choosing neural architecture and algorithms that can get the right visual representation.  In the current paper, we assume the behaviors are given, and we focus on the design choice for the selector.  We defer the question of how to  automatically develop new behaviors to include in the library to our future work. We have included this in our Limitations and Future Work section (section B in supplementary material).
>
> > Comparison to SOTA grasping behaviors from 2D images
>
> Thank you for pointing us to these related works, **we have included these in Supplementary C.2 in a more extensive literature review**. However, we do want to clarify that [11] does not show they can generalize to varying camera viewpoints, and [10] is focused on reaching tasks, not grasping. [11] is constrained to a top-down camera view from a fixed over-the-shoulder camera. [10] has not shown their grasping behavior is achieving SOTA results. Both works require significantly more data and computation compared to the proposed model. In this rebuttal, we include 2 strong baselines in grasping, Dexnet and 6D-GraspNet (please see the "Stronger baselines" section in “Response to Area Chair Fq3u”).
>
> > Design choice for grasping behaviors
>
> You are right that our current grasping behaviors are grasps from a discretized set of yaw and grasp elevation parameters. Adding more complex behaviors, and automatically detecting missing behaviors and learning them to include in the library are clear avenues for future work. **We have included these points in our Limitations and Future Work section (section B in the supplementary material).**

---

> > ### Author Response · Authors · 2021-08-30
> > **Citations in "Response to Reviewer XqcZ"**
> >
> > [1] A. Zeng, P. Florence, J. Tompson, S. Welker, J. Chien, M. Attarian, T. Armstrong, I. Krasin, D. Duong, V. Sindhwani, and J. Lee.  Transporter networks: Rearranging the visual world for robotic manipulation. CoRR, abs/2010.14406, 2020.
> >
> > [2] D. Gandhi, L. Pinto, and A. Gupta.  Learning to fly by crashing. In 2017 IEEE/RSJ International Conference on Intelligent Robots and Systems (IROS), pages 3948–3955. IEEE, 2017.
> >
> > [3] L. Pinto and A. Gupta. Supersizing self-supervision: Learning to grasp from 50k tries and 700 robot hours. CoRR, abs/1509.06825, 2015.
> >
> > [4] A. Zeng, S. Song, J. Lee, A. Rodriguez, and T. Funkhouser. Tossingbot: Learning to throw arbitrary objects with residual physics. 2019.
> >
> > [5] J. Mahler, M. Matl, V. Satish, M. Danielczuk, B. DeRose, S. McKinley, and K. Goldberg. Learning ambidextrous robot grasping policies. Science Robotics, 4(26):eaau4984, 2019.
> >
> > [6] D. C. Bentivegna, G. Cheng, and C. G. Atkeson. Learning from observation and from practice using behavioral primitives. In P. Dario and R. Chatila, editors, Robotics Research, The Eleventh International Symposium, ISRR, October 19-22, 2003, Siena, Italy, volume 15 of Springer Tracts in Advanced Robotics, pages 551–560. Springer, 2003.
> >
> > [7] R. Rahmatizadeh, P. Abolghasemi, L. Bölöni, and S. Levine. Vision-based multi-task manipulation for inexpensive robots using end-to-end learning from demonstration.
> >
> > [8] Y. Lee, J. Yang, and J. J. Lim. Learning to coordinate manipulation skills via skill behavior diversification. In International Conference on Learning Representations, 2020.
> >
> > [9] K. Fang, Y. Zhu, S. Savarese, and L. Fei-Fei. Discovering generalizable skills via automated generation of diverse tasks. CoRR, abs/2106.13935, 2021.
> >
> > [10] F. Sadeghi, A. Toshev, E. Jang, and S. Levine. Sim2real view invariant visual servoing by recurrent control.
> >
> > [11] D. Kalashnikov, A. Irpan, P. Pastor, J. Ibarz, A. Herzog, E. Jang, D. Quillen, E. Holly, M. Kalakrishnan, V. Vanhoucke, and S. Levine. Qt-opt: Scalable deep reinforcement learning for vision-based robotic manipulation.

---

### Official Review · Reviewer_UjV5 · 2021-07-23

**Originality:** Very Good
**Technical Quality:** Good
**Clarity Of Presentation:** Very Good
**Impact:** 3

**Recommendation:**

Weak Accept: I recommend accepting the paper, but will not argue for my recommendation if the majority of other reviewers have a different opinion.

**Summary:**

This works re-factors the problem of manipulation of diverse objects to building a diverse set of policies and building a data-driven policy selector. They show significantly better results on unsee objects than baselines without such a re-factor. Ablations show the importance of the view-predictive and affordance-aware representations

**Issues:**

* Include a baseline for the real robot result.
* Some discussion/analysis of why the sim baselines in Table 1 are so poor
* Discussion about when no appropriate behavior is in the library, does the selector give you a signal about this?

**Reviewer Expertise:**

Good: General knowledge of the area

**Strengths And Weaknesses:**

Strengths:
* Approach seems very data efficient and seems to generalize to new objects much better than the baselines.
* This approach may include a variety of policy types: learned or rule based.

Weakness:
* There is no baseline number reported for the real robot result, it seems like a challenging set of tasks and some baseline here would be more compelling.
* This work advocates for building a library of behaviors including new controllers, it would been nice to see some discussion around the trade-off between building these libraries versus just collecting more data for the single-policy approach.
* The interaction label process seems tedious, are there any heuristics used? For example, don't use pick and place if the task requires a pour. Are the baselines also retrained with these interaction label data in some way.
* It is surprising that the baselines perform so poorly in Table1. What are the success rates on the train objects?
* Were there any cases where this approach can tell you if you are missing a behavior to tackle the task?
* How were the real robot controllers designed? Did you have knowledge of the test objects during that design?


**Summary Of Recommendation:**

In some limited settings, this approach seems very efficient for manipulation tasks, however, it is not clear how generalizable the result is. Stronger baselines on real robot results would have made this work more compelling. Additionally, some discussion of the trade-offs of having to develop and maintain libraries of policies and the additional cost of the interaction labeling versus single-policy approach with more data.

---

> ### Author Response · Authors · 2021-08-30
> **Response to Reviewer UjV5 (1/2)**
>
> Thank you for your valuable comments. Here we address your concerns. We have also updated the paper according to your suggestions (bold-faced in the reply).
>
> > Quantitative results on real robot
>
> (Please see the “Quantitative results on real robot” section in “Response to Area Chair Fq3u”.)
>
> > Building libraries versus collecting more data for the single-policy approach
>
> Thanks for your suggestion, **we have included a discussion in Supplementary C.1**.
>
> To improve the performance and capacity of a single image-to-action policy, the cost is not only about collecting more data, it is also about scaling up the size of your models, computation, training schema and parameter search. Unfortunately, this approach often needs an immense amount of data and computation to train [1, 2], which most research groups do not have access to.  Another direction is to put engineering effort in designing specialized architectures or representations for your application, e.g., the DexNet architecture is specialized for top-down grasp. We see this type of approach can achieve better results compared to the end-to-end approach with deep neural networks. However, it is often restricted to the specific domain it is designed for.
>
> For the library of behaviors approach, the main effort is in making the behaviors. Fortunately, when designing new behaviors for the library approach, there is no restriction on the input-output the model should use and the algorithm to obtain them, which makes the engineering work relatively simple. Recent advances in deep RL and large-scale imitation learning from human demonstration will be a promising approach to scale up the behaviors with domain experts. Since any behavior, no matter learned or engineered, can be put in the library, we can potentially combine the efforts from both traditional robot controls and modern deep learning approaches.
> Another key issue to consider is how one can update the model to include a new set of skills. For the single-policy approach, one will need to retrain the whole system with new data, and it is not guaranteed that tasks that have been tackled before can be successfully tackled after the retraining. On the contrary, explicitly maintaining a library of behavior makes it simple to incrementally increase a new set of skills without breaking the existing ones. The only thing that needs to retrain is the selector, which is relatively simple to train.
>
> > Heuristics used in getting the interaction label
>
> You are right that in real robot experiments we did use some heuristics to speed up data collection. **We have included a description in Supplementary F.3. to ensure this is clear.** When asked to pour something, we label failure for all grasping controllers because they do not perform the pouring action; also when grasping long objects, we label obviously infeasible grasping angles as failures.
>
> > Training baselines with interaction labels
>
> Thanks for pointing out, we did train our baselines with interaction data. **We have included the details in section 4.4.** In behavior cloning and DDPG-HER, we collect a set of demonstrations with equivalent size compared to the interaction label data to train the agent. These models need to learn from demonstration trajectories as opposed to binary interaction labels.
>
> > Poor baseline performance
>
> Thank you for pointing this out, **we have included the discussion in Supplementary A.3**.  The baselines have similar performances on the training and testing data, meaning these models are underfitting. The Abstract 3D baselines do not perform well because they do not have visual input and lack information about the object’s shape; the Abstract 3D + Image do not perform well because it operates in 2D image space where the representation (of objects, their poses, and appearance) can change dramatically due to camera pose change, making the learning problem difficult; the Contextual 3D baselines need much more compute due to the 4D bottleneck and do not learn even with strong supervision from imitation learning. For all the baselines, we have grid-searched for the best parameters used in the model, as mentioned in L261. Models with more parameters are in general more difficult to train and do not necessarily give better results.
>
> > Detecting missing behaviors to tackle a task
>
> Thanks for your suggestion! Our current framework doesn't detect missing behavior or automatically learns the missing behavior, but this is clearly a key avenue to explore in the future! **We have included it in our Limitations and Future Work section (section B in supplementary material).** With the current model, you can manually analyze failure objects for the trained system and intentionally add behaviors that can solve these failure objects. An advantage of our framework is that you can easily include new behaviors without "forgetting" old behaviors.

---

> > ### Author Response · Authors · 2021-08-30
> > **Response to Reviewer UjV5 (2/2)**
> >
> > > Design choice for the real robot controllers
> >
> > We build the library of behaviors in the real robot experiment by asking the questions “what behaviors would be needed to plate different kinds of food on a plate?” We then pick several representative training objects and build the placing and pouring controllers from them. We know the testing objects might be in the same category as some of the training data, but we do not assume we know their shapes, appearance, and textures. Please see Supplementary F.2 for more details.
> >
> > _Citations:_
> >
> > [1] S. Levine, P. Pastor, A. Krizhevsky, J. Ibarz, and D. Quillen. Learning hand-eye coordination for robotic grasping with deep learning and large-scale data collection. The International Journal of Robotics Research, 37(4-5):421–436, 2018.
> >
> > [2] F. Sadeghi, A. Toshev, E. Jang, and S. Levine. Sim2real view invariant visual servoing by recurrent control.

---

### Official Review · Reviewer_GybB · 2021-07-25

**Originality:** Good
**Technical Quality:** Good
**Clarity Of Presentation:** Good
**Impact:** 3

**Recommendation:**

Weak Accept: I recommend accepting the paper, but will not argue for my recommendation if the majority of other reviewers have a different opinion.

**Summary:**

The main contribution of this paper is a novel pipeline for learning grasping and pushing actions that integrates both raw sensory data as well as more abstract object features.


**Issues:**


In section 3, the 3D bounding box would work for relatively simple objects in which the bounding box would have little else but the object inside of it; what about more complicated objects? Also, is the bounding box aligned with the principal axis of the object if there is one?

In section 3.1 when describing the training, is the randomly chosen behavior still condition on the goal for the particular trial? Also, at this point in the paper it is not clear how do you deal with behavior parametrizations, e.g., push can be parametrized with force, direction, etc?

With regards to the 3D object detector, is this learned for each individual object or do you learn a detector that works on all objects, even ones not seen at training? Also, were the test objects in the experiments excluded from the training data for the object detector?

Section 3.2, line 193, why not also change in orientation of the end effector when it comes to the action space? Currently it is stated that it can only change its position (translation) but not orientation. Ideally, the dimensionality of the action space should be stated explicitly.

Section 3.2, line 195, needs more detail here -- what is the difference between the 25? just the goal? also, how are the objects split into train and test and why is one behavior trained on all objects?

Similarly, for grasping, it not at all clear where these 30 grasping controllers come from and also it is not clear how the grasping point relative to the center of the object is actually selected (or who selects it)

In section 4.1, there needs to be more detail on the simulation environment, in particular, how does it model things like texture, friction, etc.? Are all the objects assumed to have the same material and density and only the shape is different?

With regards to object augmentation in line 228, does this mean that on average, from each original training object, you get 3 augmented objects and configurations?

With regards to the sentence, "An episode is successful if the object centroid is within 5cm of the target at the final timestep.", how is it decided when an episode ends, is there a fixed limit?

With regards to how objects are split into train and test, do the categories play a role e.g., do you ensure that the test set only has objects from novel categories or do you ensure that for each category, some objects are in train and some in test?

For the grasping pick and place task, it would be nice to have a comparison of a traditional approach, like Agile grasp or some other classifier used for point selection and then planning to move it to a target location -- at least, what are your thoughts on how your system would compare against a more traditional approach? The current baseline comparisons are OK but in some ways, it feels like there were selected because they were doomed to fail (there is also the whole issue of how hyperparamters are selected for your approach vs baselines and without any principled methodology for doing so, I can only assume that the graduate students spent any time tuning any parameters mostly for the proposed approach)


some minor issues:

figure 3 could use a longer caption

line 63 of the intro and at other places talk about the object being "diverse" but it is not explained, diverse with respect to what? all the objects seem mostly rigid, simple shapes so what are they diverse in terms of?

line 7 in the abstract, the phrase "in hand," is a bit confusing as the object is not always in he robot's hand

The paper often puts "object location and pose" next to each other, but pose already includes location information so it's a bit redundant.


**Reviewer Expertise:**

Very good: Comprehensive knowledge of the area

**Strengths And Weaknesses:**


The main strength of the paper in my opinion are the results with the real robot (personally, I would focused mostly on these as opposed to the results in simulation)

The main weakness of the paper is the comparison against a weak baseline, almost designed to fail. There clearly are end-to-end systems designed for grasping and pushing that in theory should be able to achieve better results then the baselines (though I understand that training such systems is an art in itself and how hypermarkets are decided can have big impact on the results). The other main weakness of the paper is the lack of many details without which it is a bit difficult to judge the level of contribution (see issues later in the review).

**Summary Of Recommendation:**

Overall, the paper brings in some new ideas and the evaluation with a real robot is great. There are still many issue to be addressed and if there was a "borderline" option between weak accept and weak reject, I would have picked that one.

---

> ### Author Response · Authors · 2021-08-30
> **Response to Reviewer GybB**
>
> Thank you for your detailed review. It seems the two main concerns are regarding the baselines and clarifications about some experimental details. We have edited the paper to include the baselines and more details on the unclear parts. Here we address your concerns.
>
> > Stronger Grasping and Pushing baselines.
>
> (Please see the “Stronger baselines” section in “Response to Area Chair Fq3u”.)
>
> > Details and Clarifications.
>
> 1. **“Is the bounding box aligned with the principal axis of the object if there is one? How to handle more complex objects?”** Since we assume the object category is unknown, objects do not have a canonical pose (principal axis). The 3D bounding boxes are aligned to the axes in the world coordinate system (“axis-aligned” in L164). Complex objects might need multiple bounding boxes to capture different parts of the object, and the selector would need to compare several parts between two objects. Extending the current framework to handle objects with multiple parts is an interesting direction for future work (see L84 in supplementary material).
> 2. **Behavior parametrizations.** All our behaviors are goal conditioned, no matter whether it is randomly chosen or selected by the learned selector. A pushing behavior is parameterized by the 3D translation (direction), and the opening/closing state of the gripper, as specified in L193. The total dimension is four. We did not include force in the parameterization. A grasping behavior is parameterized by the 3D grasping point and angles, as specified in L200. The total dimension is five.
> 3. **“Is the detector learned for each individual object or do you learn a detector that works on all objects? Do you exclude the testing objects when training the detector?”** The detector is category agnostic, meaning it can capture any unseen object without knowing its category. Yes, we only use the training objects to train the detector (L169).
> 4. **“Why not also change the orientation of the end effector in the pushing behavior?”** This is a good suggestion. More complex behaviors are future work. What we observe is that our current pushing behaviors can already push the object to a target even when the path to the target is not in a straight line, because the model can link several translational pushes instead of making the behavior more complex. But we agree that including more complex pushing behaviors will allow the agent to achieve more complex tasks.
> 5. **Training details for obtaining the 25 pushing behaviors.** Among the 25 behaviors, one of them is an RL policy trained on all objects. For the rest of the 24 behaviors, we train policies on disjoint subsets of training objects organized based on object category and initial poses, as mentioned in L196. We include one behavior trained on all objects because that behavior can already solve a reasonable amount of objects and configurations so the rest of the behaviors can focus on specific objects and configurations that this trained-on-all behavior cannot solve. To split train and test sets, we split objects in every category into train and test sets (see section D of supplementary materials for more details)
> 6. **How to determine the parameters for the 30 grasping behaviors.** We aim to select grasping points and angles that uniformly surround the target objects and can potentially grasp the objects. We do this by designing top-down center grasps with different gripper orientations, top-down side-grasp that tries to grasp different parts on an object that is close to the boundary of the object, and side-grasp that tries to grasp the boundary with a tilted gripper. More details can be found in Table 2 in the supplementary materials.
> 7. **Material and density of the simulated objects.** In our experiments, the materials and densities of all objects are identical. We have included the details in section 4.1.
> 8. **“Do you get 3 augmented objects and configurations from a training object (mesh)?”** Yes, that’s correct.
> 9. **“How to decide when an episode ends, is there a fixed limit?”** Our agent runs in an environment with a fixed episode horizon of 50 timesteps (L241); after the episode ends, we measure the distance between the object and the target to determine whether it succeeds or not.
> 10. **Comparison to SOTA grasping approaches.** Please see the “Stronger baselines” section in “Response to Area Chair Fq3u”.
> 11. **Diversity in objects.** The objects have large variations in their shapes, e.g., some are solid while some have an empty hole in the middle; some are tall while some are flat. The variations make the pushing and grasping manipulation especially challenging.

---

### Official Review · Reviewer_7MZX · 2021-07-25

**Originality:** Very Good
**Technical Quality:** Excellent
**Clarity Of Presentation:** Excellent
**Impact:** 4

**Recommendation:**

Weak Accept: I recommend accepting the paper, but will not argue for my recommendation if the majority of other reviewers have a different opinion.

**Summary:**

The paper proposes a two stage hierarchical approach for learning to manipulate objects in a way that generalizes across objects, configurations, and views. The lower level consists of a library of learned or engineered behaviors that can handle a subset of these variations. These behaviors then become discrete actions of a higher level policy, which operates on an estimated 3D scene representation, transformed into the robot frame. The scene representation is then cropped and mapped onto a latent vector representation that is jointly learned with a vector representation for each behavior. At inference time, the behavior is chosen based on the similarity of these vectors. The method is evaluated in simulated and on a real robot in grasping, pushing, and transport tasks.

**Issues:**

If possible, additional evaluations as discussed above would strengthen the paper.


**Reviewer Expertise:**

Good: General knowledge of the area

**Strengths And Weaknesses:**

The proposed method combines a variety of deep learning machinery and models in a structured way. The idea of having a two-stage approach to generating behavior is not new but the way how this is done here is very original and there are many ideas in there that make a lot of sense, e.g. jointly training the input and behavior representations, making generalization over viewpoints explicit via self-supervised objectives and via estimating the scene representation in 3D and transforming it into the robot frame.

This already good paper could be further improved by extending the experimental section with qualitative results that visualize the components of the system. How good are the learned 3D representations in sim and in real? What does the embedding space of the behaviors look like? What are the typical failure cases of the proposed method?


**Summary Of Recommendation:**

This is a nice paper that includes a lot of relevant ideas that should be interesting to the field. The paper appears correct and significant to me. It includes experiments in simulation and on a real robot. The experimental evaluation could be improved with additional analysis (see above).

---

> ### Author Response · Authors · 2021-08-30
> **Response to Reviewer 7MZX**
>
> We thank the reviewer for your thoughtful comments. We are encouraged to hear that you appreciate the ideas and find the paper enjoyable to read. We appreciate your suggestions about including more qualitative results regarding the learned representations and the failure case. We have also updated the paper according to your suggestions (bold-faced in the reply).
>
> > Visualization for the learned representation
>
> **We visualize our visual feature space in Supplementary A.5 with t-SNE projection of the learned embedding space.** From the t-SNE projection, we can see that objects that have similar affordances are grouped together in clusters and the corresponding behaviors that can manipulate these objects well are also encoded at the corresponding positions in the latent space. You can see some objects in different categories get assigned to the same controller since they have similar affordances.
>
> > Failure cases of the proposed method
>
> Our method typically fails when (1) the selector makes the incorrect prediction, e.g., grasping a cucumber with the wrong grasping angle (please see 2:13 of the supplementary video for the example) and (2) when all the behaviors in the library cannot successfully operate the target object, e.g., when trying to pour grapes from a container, although the trained selector chooses the correct pouring behaviors, the grapes did not successfully get out of the container due to strong friction (see 3:31 of the supplementary video for the example). The first issue can be solved by improving the 3D feature representations. One way is to improve the 3D resolution with memory-efficient structures such as point clouds. Another way is to improve the features using recent advances in self-supervised feature learning. The second issue can be improved by adding behaviors to the library or automatically adapting existing behaviors in the library. Both are interesting future avenues to explore. **We have included the discussion in Supplementary A.4 and B.**

---

### Meta-Review · Area_Chair_Fq3u · 2021-08-11

**Recommendation:** Accept (Poster)
**Confidence:** 3

**Metareview:**

Strengths:

- The view invariance of the proposed method (a result of the use of GRNNs) is a key strength and seems to generate over views better than the baselines.

- The high level structure of the approach where the perceptual module and the behavior keys are trained jointly was viewed as a strength. The fact that the low level policies could be learned or rule-based was also viewed as a strength.

Weaknesses:

- Two reviewers asked for stronger baselines. This is in reference to Table 1 where the proposed method is evaluated on a class of grasping tasks and a class of pushing tasks. The baseline methods perform terribly on these tasks and on grasping in particular. However, there are lots of grasp control/detection methods available now that can do very well with grasping, especially in the single-object scenario that we consider here. There are also a variety of push control methods that can similarly do well.

- There was a concern that the fixed set of behaviors made the approach overly simplistic. Essentially, in the real robot exps at least, the agent looks at the scene and then selects one of a fixed set of motion scripts. Joint learning of behavior policies could help here. Another approach might be to interpolate amongst learned skills at test time.

- Multiple reviewers asked for more comprehensive real robot exps.

Post Rebuttal:

The authors added a significant amount of content and comparisons into the appendix (supplemental), including grasp comparisons which we appreciate, as well as the t-sne plot which was cool. However, if this work is viewed through the lens of "k-way" classifier over pre-scripted behaviors, then (as the authors point out) the remaining contribution is the space in which the scene is represented. Possibly this is really prior work rather than a new contribution here. Nevertheless, the application to manipulation goes beyond prior work and was viewed positively.

---

> ### Author Response · Authors · 2021-08-30
> **Response to Area Chair Fq3u**
>
> Thank you for summarizing the strengths and weaknesses of our work as well as the main points of the initial reviews! We address the detailed questions from the reviewers in responses to each reviewer below, while here we respond to high-level points made in the meta-review. We have also updated the paper according to the reviewers’ suggestions (bold-faced in the reply).
>
> > Stronger baselines
>
> Following the suggestions from the reviewers, **we have added evaluation results on state-of-the-art grasping methods, 6D-GraspNet and DexNet, and the discussion in the supplementary section A.2**. We have fine-tuned both models with the same amount of interactive labels as our model. The success rates on unseen objects are 0.36 and 0.72, respectively, which are worse than that of the proposed model (0.78). We found 6D-GraspNet performs much worse than what is reported in the original paper under our setup, and we attribute this to the fact that our setup is more difficult since we randomize the initial and goal locations, and camera poses of the objects. We found 6D-GraspNet sensitive to camera pose change. Many proposed grasps turn out to be unstable when an object is placed too close or far away from the robot. DexNet performs reasonably well on objects that can be grasped with a top-down grasp, but fails completely on objects that require a side-grasp, e.g., plates.
>
> > Quantitative results on real robot
>
> We thank the reviewers for the suggestions. **We have added two baselines on our real robot experiment in section 4.4** to emphasize the importance of the proposed hierarchical structure with 3D representations: 1) an image-to-action model trained with behavior cloning (image), and 2) a hierarchical model that uses a library of behaviors and a selector with 2D representations (2D features), as suggested by the reviewers. For fair comparison, we use the data collected during the interaction label collecting process as the data used to train our baseline models. We test the models on the plating task. For the Image baseline, we are not able to obtain appreciable results, while we did make it work in simulation where there is more data. For the 2D-features baseline, we get a success rate of 38.0%, which is significantly worse than that of the proposed model (88.6%). This again shows the importance of operating the selector in the proposed view-invariant and object-centric 3D feature space.
>
> > The fixed set of behaviors made the approach overly simplistic
>
> If the concern is that the set of behaviors is fixed, we look forward to exploring how to add new behaviors as future work, and **we have included more discussions about this issue in the Limitations and Future Work section (section B of the supplementary materials)**. This current paper focuses on learning from pre-existed behaviors, and not on inventing the behaviors on the fly. Our hypothesis is that humans use a vast library of behaviors, and most of them are not invented from scratch. Anecdotal evidence is along the following lines: “how do you complete task X (in this case, opening a jar where the metal lid is stuck)?” Typical answers are “put a cloth on the top” or “run hot water on the top”. When you are asked “Did you invent this behavior”, the answer is typically “No, I learned it from watching others [learning from observation] or being taught [learning from demonstration and coaching]”. Inspired by this observation, we hope to scale up the library of behaviours to solve diverse tasks by leveraging recent advances in large-scale imitation learning from human demonstration [1].
>
> [1] S. Young, D. Gandhi, S. Tulsiani, A. Gupta, P. Abbeel, and L. Pinto. Visual imitation made easy, 2020.
>
> > Joint learning of behavior policies
>
> We appreciate your suggestions that adopting a joint learning scheme will allow the model to develop skills that can cover task space that are not readily covered by the behaviors in the library. However, here we do want to clarify that joint learning, possibly from scratch as suggested by existing work, is in general very difficult to learn since: a) learning a policy from scratch through RL is hard (we also have shown this in the paper), b) learning a number of policies and a selector at the same time will make the learning problem even harder, and c) the learned policies can be more restrictive than the proposed model in the sense that these jointly learned policies need the same input representation. We agree that developing skills beyond the capacity supported by a fixed set of behavior is critical, and here we defer the question of how to automatically develop new behaviors to include in the library as a clear avenue for future work. **We have included this discussion in the Limitations and Future Work section (section B in the supplementary material).**

---

### Decision · Program_Chairs · 2021-09-13

**Decision:**

Accept (Poster)

**Comment:**

Strengths:

- The view invariance of the proposed method (a result of the use of GRNNs) is a key strength and seems to generate over views better than the baselines.

- The high level structure of the approach where the perceptual module and the behavior keys are trained jointly was viewed as a strength. The fact that the low level policies could be learned or rule-based was also viewed as a strength.

Weaknesses:

- Two reviewers asked for stronger baselines. This is in reference to Table 1 where the proposed method is evaluated on a class of grasping tasks and a class of pushing tasks. The baseline methods perform terribly on these tasks and on grasping in particular. However, there are lots of grasp control/detection methods available now that can do very well with grasping, especially in the single-object scenario that we consider here. There are also a variety of push control methods that can similarly do well.

- There was a concern that the fixed set of behaviors made the approach overly simplistic. Essentially, in the real robot exps at least, the agent looks at the scene and then selects one of a fixed set of motion scripts. Joint learning of behavior policies could help here. Another approach might be to interpolate amongst learned skills at test time.

- Multiple reviewers asked for more comprehensive real robot exps.

Post Rebuttal:

The authors added a significant amount of content and comparisons into the appendix (supplemental), including grasp comparisons which we appreciate, as well as the t-sne plot which was cool. However, if this work is viewed through the lens of "k-way" classifier over pre-scripted behaviors, then (as the authors point out) the remaining contribution is the space in which the scene is represented. Possibly this is really prior work rather than a new contribution here. Nevertheless, the application to manipulation goes beyond prior work and was viewed positively.